# The Effectiveness of eHealth Interventions in Promoting the Health Literacy of Informal Caregivers: A Systematic Literature Review Protocol

**DOI:** 10.3390/healthcare12232354

**Published:** 2024-11-25

**Authors:** Patrícia Valentim, Paulo Costa, Adriana Henriques, Paulo Nogueira, Andreia Costa

**Affiliations:** 1Nursing Research, Innovation and Development Centre of Lisbon (CIDNUR), Nursing School of Lisbon, Avenida Prof. Egas Moniz, 1600-190 Lisbon, Portugal; p.costa@esel.pt (P.C.); ahenriques@esel.pt (A.H.); pnogueira@medicina.ulisboa.pt (P.N.); andreia.costa@esel.pt (A.C.); 2Instituto de Saúde Ambiental (ISAMB), Faculdade de Medicina, Universidade de Lisboa, Av. Prof. Egas Moniz, Ed. Egas Moniz, Piso 0, Ala C, 1649-028 Lisboa, Portugal; 3Laboratório para a Sustentabilidade do Uso da Terra e dos Serviços dos Ecossistemas—TERRA, 1349-017 Lisboa, Portugal

**Keywords:** digital health interventions, health literacy, informal caregivers, nursing

## Abstract

Background: Digital Information and Communication Technologies (ICTs) offer significant opportunities to enhance public health, particularly through their strategic use in promoting health literacy. Objectives: This systematic review protocol aims to outline the methodological steps necessary to conduct a systematic review of the effectiveness of digital interventions in improving health literacy among informal caregivers. Methods: This review will include studies involving adult informal caregivers (≥18 years) undergoing a digital health intervention promoting health literacy. Intervention, effectiveness or efficacy, RCT, quasi-experimental, and observational studies will be eligible. This review will follow the methodology for Cochrane Systematic Review and Meta-Analysis. The search strategy will identify studies published in the databases SCOPUS and Web of Science, as well as CINAHL (via EBSCO) and PubMed and gray literature sources. Two in-dependent reviewers will screen the studies, extract data, and critically appraise the selected studies. It is planned that the risk of bias will be assessed using the RoB2, Newcastle-Ottawa Scale (NOS), and the ROBINS-I (Risk Of Bias In Non-randomized Studies of Interventions). From the included studies, data will be extracted on the identification of the study, the aim of the study, characteristics of the population, study method and intervention and control conditions, study variables, and significant results. Conclusions: It is anticipated that the results of the study will guide healthcare professionals and managers in incorporating digital technologies into health literacy programs for informal caregivers. Whenever possible, a statistical meta-analysis will be performed to combine study results. The PROSPERO registration number is CRD42024589465.

## 1. Introduction

Health literacy is a fundamental concept in global health and is crucial for achieving both individual and collective health objectives [1,2,3,4,5]. Considering health literacy involves acknowledging a person’s life journey in a holistic approach, where health policies must be strategic and responsive to people’s needs, promoting health-favorable behaviors [4,6,7,8]. The World Health Organization (WHO) emphasizes the importance of health literacy in maintaining health through a set of cognitive and social skills and the ability to access, comprehend, and use health information. It highlights the significance of people’s knowledge, motivation, and skills for proper understanding and utilization of information [9].

Digital technologies present new opportunities and challenges for achieving health objectives, and their strategic use will be crucial in ensuring that more people benefit from universal health coverage, are protected against health emergencies, and enjoy better health and well-being. In the 2020–2025 global digital health strategy, the WHO defines digital health as the domain of knowledge and practice associated with the development and use of digital technologies to improve health. The digital transformation of health can be disruptive, yet technologies such as the Internet of Things, virtual care, remote monitoring, artificial intelligence, big data analysis, digital platforms, and the sharing of relevant data in the healthcare ecosystem enable care continuity, better health outcomes, more accurate medical diagnoses, self-management of care, and person-centered care, as well as more evidence-based knowledge, skills, and abilities [10]. The WHO recognizes the potential of digital health to achieve Sustainable Development Goals, particularly by supporting health promotion and disease prevention, improving the quality and accessibility of health services worldwide [11].

The literature indicates that populations with higher levels of digital literacy exhibit positive indicators, such as adopting behaviors that promote health and prevent diseases, greater psychological well-being, a critical sense towards their health, and more efficient use of health services [12]. Conversely, low levels of health literacy are linked to a reduced quality of life, leading to a higher number of hospitalizations, more frequent use of emergency services, and a lower prevalence of individual and family preventive attitudes [13]. Vulnerable groups, such as people aged 65 and older, those with low education levels, and individuals with chronic diseases, are particularly affected [13,14].

Advances in science and technology and the increasing prevalence of Non-Communicable Chronic Diseases (NCDs) also result in longer lifespans but with fewer disease-free years [1,15] and an increase in the number of informal caregivers. International studies report an association between higher health literacy levels among informal caregivers and their ability to access support services. These are also linked to reduced caregiver burden, better health-related quality of life, and less time spent on caregiving tasks [16,17]. It is, therefore, essential to improve health literacy by involving caregivers in initiatives that enhance access to, comprehension of, evaluation of, and use of health information [1] to promote healthy lifestyles [6]. In this sense, developing studies on health literacy is crucial to positively impact the role performance of informal caregivers within healthcare systems through digital health interventions.

An initial search was conducted to test the feasibility of the study topic, and several studies were identified, including a 2023 Systematic Literature Review and Meta-Analysis, which aimed to understand the effectiveness of digital health literacy interventions in adults. Seven studies were included, concluding that digital health literacy interventions positively impact older adults’ health status and management. The study highlights the relevance of practical and effective digital literacy interventions for older adults’ health management but notes that, due to variability, the effectiveness of these interventions should be further explored, especially in other populations like informal caregivers [18]. Another study, a Scoping Review, mapped scientific evidence from nine studies on the impact of digital technologies in promoting health literacy and empowering informal caregivers [19]. It found that informal caregivers who use digital tools such as computers, smartphones, and the internet to access information, manage household tasks, and communicate with family and healthcare professionals benefit from their use, enhancing their quality of life. Integrating multiple resources into a technological support tool for care saves time and facilitates daily tasks. However, there are few studies investigating the impact of these digital interventions specifically on promoting health literacy among informal caregivers. Therefore, it is crucial to understand the effectiveness of digital health interventions in promoting health literacy and identify which are most effective and beneficial for caregiver empowerment [13]. Another Scoping Review explores the use of digital health technology to assist healthcare professionals and family caregivers in caring for individuals with cognitive deficits living in the community [20], and it noted that research on this topic is scarce, recent, and heterogeneous, highlighting the need for a theoretical framework that incorporates digital technology.

In this context, this systematic review is expected to contribute to a theoretical basis for future research to fill a significant knowledge gap on the use of digital health interventions that promote health literacy among informal caregivers. eHealth interventions, such as care support apps and patient health portals, have been widely adopted to facilitate health management. However, the effectiveness of these tools in promoting health literacy among informal caregivers has not yet been systematically evaluated. This systematic review is, therefore, crucial to synthesizing evidence on how such interventions can be optimized to support caregivers, particularly given the increasing reliance on digital solutions in healthcare. Thus, there is a need to promote the research and development of strategies focused on digital health interventions that enhance health literacy among informal caregivers. Current studies provide a limited view of the impact of digital interventions on informal caregivers, emphasizing the need for further research in this area to promote the development of efficient care and reduce research gaps globally. Consequently, the following research question was formulated: What is the effectiveness of digital health interventions in promoting health literacy among informal caregivers?

Conducting this study will be essential to advancing scientific knowledge, helping to identify the most effective digital interventions in promoting health literacy and empowering informal caregivers, thereby improving their quality of life and well-being.

## 2. Materials and Methods

This protocol was structured according to the Preferred Reporting Items for Systematic reviews and Meta-Analyses for Protocols (PRISMA-P) [21,22] statement, which is available in the Appendix A Archive. The systematic review will be conducted according to the guidelines of the Cochrane Collaboration Handbook for Systematic Reviews of Interventions [23]. The protocol is registered in the international database of systematic reviews: PROSPERO (registration number CRD42024589465).

### 2.1. Inclusion Criteria

#### 2.1.1. Participants

In terms of participants, this review will cover studies that focus on adult informal caregivers (≥18 years) who are characterized by being a person (e.g., spouse, partner, family member, friend, or neighbor) who provides unpaid care to someone with a chronic illness, disability or other long-term health need, outside of a professional or formal setting [24].

#### 2.1.2. Intervention

This review will include studies that address digital interventions (such as apps and websites) aimed at promoting health literacy, specifically the development of an app that has contributed to increasing health literacy among informal caregivers. In this context, the intervention to be investigated refers to eHealth solutions focused on promoting health literacy among informal caregivers.

#### 2.1.3. Comparator

This review will include studies that evaluate digital interventions, with or without comparators, regardless of the type of digital intervention.

#### 2.1.4. Outcomes

The review aims to provide a comprehensive assessment of the results that have been selected regarding health literacy and, more specifically, of the studies that used methods to quantitatively measure health literacy, namely, through the Health Literacy Scale (HLS), HLS-EU-Q, eHEALS, among others. The effectiveness of an eHealth intervention in promoting health literacy (HL) will be measured based on studies that used quantitative methods to assess health literacy, specifically through a tool designed to evaluate health literacy.

#### 2.1.5. Limitation on the Types of Study

This review will include intervention, efficacy or effectiveness, randomized controlled trial (RCT), quasi-experimental, and observational study designs. Excluded studies are descriptive studies, qualitative studies, case reports, expert opinions, editorials, commentaries, conference abstracts, dissertations, protocols, and literature reviews. Any studies that are not available in full-text format will also be excluded, as well as articles published more than 10 years ago. Studies available in English and Portuguese will be included due to linguistic comprehension.

### 2.2. Search Strategy

We will employ a three-step search strategy in this systematic review. In the first step, we will conduct an initial, limited search of at least one relevant online database, such as SCOPUS. This initial search will help us to identify relevant studies and allow us to analyze the keywords, text words in the titles and abstracts, and index terms used to describe the retrieved articles. In the second step, we will conduct a comprehensive search across all selected databases using the keywords and index terms identified during the initial search. This approach will ensure that we capture all potentially relevant studies. Finally, in the third step, we will review the reference lists of the identified reports and articles to find additional sources that may not have been captured in the database searches. If necessary, we will contact the authors of primary studies via email to obtain further information.

To carry out a comprehensive systematic review of the literature, we will submit queries to the databases SCOPUS and Web of Science, as well as CINAHL (via EBSCO) and PubMed. Gray literacy will also be included for analysis.

The search strategy for the SCOPUS database is presented as a detailed example in the Appendix A File. For other databases, the search strategy will be adapted.

The evidence-based Peer Review of Electronic Search Strategies (PRESS) guidelines for systematic reviews will be followed [25].

### 2.3. Study Selection

The search results from the databases will be exported to Rayyan^®^ (https://www.rayyan.ai/, accessed on 5 September 2024), where duplicates will be identified using the software’s automatic similarity screening tool. A designated member of the research team (P.V.) will review these duplicates and make the final decision on their removal. Following this, an initial screening of the titles and abstracts will be conducted by two reviewers (P.V. and A.C.) according to the predefined eligibility criteria. Subsequently, the full text of the selected articles will be examined. In cases where the two reviewers disagree on the eligibility of an article, a third reviewer (A.H.) will be consulted to resolve the conflict. The PRISMA-S flowchart (Extension) will be utilized to document and describe the process of selecting the final studies included in the review.

### 2.4. Assessment of Methodological Quality

The risk of bias will be assessed using RoB2, a Cochrane tool recommended for evaluating the risk of bias in randomized studies. RoB2 is structured based on a fixed set of bias domains, focusing on different aspects of trial design, conduct, and reporting. Within each domain, information is collected on the characteristics of the study that are relevant to the risk of bias. The proposed risk of bias for each domain is translated into an algorithm. The result can be “Low”, “High”, or “Somewhat Unclear”, depending on the risk of bias. For each domain, it is planned that two authors, independently (P.V. and A.C.), will answer the signaling questions regarding the risk of bias. In the event of classification conflicts, both authors will discuss and reach a consensus, with the assistance of a third author. In case of disagreement, the third reviewer (A.H.) will decide on the final quality score of the study.

For observational studies, we plan to use the Newcastle-Ottawa Scale (NOS) and, for intervention studies, the ROBINS-I (Risk Of Bias In Non-randomized Studies of Interventions).

### 2.5. Data Extraction

Two authors are expected to extract the data from the studies independently, coding it in an Excel file. When the necessary information is not fully described in the article, it is planned to contact the authors of the article.

The data extracted will include specific details on the identification of the study (authors’ identification; year of publication; country; study title; study design), the aim of the study, characteristics of the population (sample size; gender; average age; educational level; characteristics of the population—patient, non-formal caregiver or non-patient), study method and intervention and control conditions (digital intervention format; duration of the digital intervention; regularity of the digital intervention; identification of the health literacy assessment scale), study variables (assessment time; study bias category) and significant results. New categories may be created during the analysis, especially if themes or categories are identified that were not initially anticipated.

The information extracted will be compared between the two reviewers. At each stage of the data extraction process, any disagreements between the reviewers will be resolved through discussion or with a third reviewer.

### 2.6. Data Synthesis

Whenever possible, the studies will be grouped together with a meta-analysis. To assess the effectiveness of digital interventions in promoting health literacy, it is planned to collect pre- and post-intervention data from the control group and the intervention group to obtain the SE and d-Cohen and then carry out the meta-analysis using the Jasp^®^ (https://jasp-stats.org/, accessed on 5 September 2024) tool. If observational studies are included in the review, the data from these studies will be grouped separately from the data from experimental or quasi-experimental studies. If the results show heterogeneity, moderation of the intervention format and subgroup analysis are planned. The subgroup analysis will be based on the variables that may influence the effect of the intervention, namely, the population, age/age group, types of health literacy assessment tools, among others. The inclusion of risk of bias as a moderator could be considered. If a study does not report its design effect or does not provide intraclass correlation coefficients, we will contact the authors to obtain the necessary information. The research team will apply funnel plot-based methods to deal with possible publication bias when there are 10 or more studies included in a meta-analysis. When meta-analysis is not possible, the results will be presented in narrative format, including tables and figures, to aid in the presentation of the data.

## 3. Discussion

eHealth interventions, such as caregiving support apps and patient health portals, have been widely adopted to facilitate health management. However, the effectiveness of these tools in promoting health literacy among informal caregivers has not yet been systematically evaluated. This systematic review is, therefore, crucial for synthesizing evidence on how such interventions can be optimized to support caregivers, especially considering the increasing reliance on digital solutions in healthcare. In this regard, this review aims to bridge that gap related to the impact of digital interventions on health literacy among informal caregivers, an area where there is a limited amount of consolidated evidence. Although eHealth technologies are rapidly expanding, it remains uncertain which types of interventions are most effective for different populations of caregivers and health contexts. This study aims to fill this gap by providing a comprehensive analysis of the most effective eHealth interventions.

The relationship between digital health interventions and the promotion of health literacy will be evaluated based on the effect of health literacy measurement scales, whether positive or negative. To assess the effectiveness of digital interventions in promoting health literacy, it is planned to collect pre- and post-intervention data from both the control group and the intervention group in order to obtain the SE and d-Cohen. If the meta-analysis is feasible, the investigation will focus on quantitative outcomes that evaluate the effectiveness of eHealth interventions in promoting health literacy among informal caregivers, measured by the increase in health literacy scores, as assessed by standardized instruments such as the Health Literacy Scale (HLS). The effect sizes included will be means, Odds Ratios (ORs), or Risk Ratios (RRs) for categorical outcomes, such as the proportion of caregivers who demonstrated an adequate level of health literacy after the intervention. Hedges’ g or Cohen’s d will be used for specific types of effect sizes to quantify the magnitude of differences between groups, particularly relevant in studies measuring changes before and after interventions.

## 4. Conclusions

This protocol presents a structured approach for conducting a systematic literature review and, if feasible, a meta-analysis, with the aim of investigating effective digital health interventions for promoting health literacy among informal caregivers. The results of this study may have significant practical and theoretical implications, opening new opportunities for future research. The findings could inform healthcare professionals and service managers about the most effective eHealth interventions to implement in order to support and empower informal caregivers in promoting health literacy. Beyond identifying effective strategies, this investigation is also expected to highlight gaps in the literature that could guide future studies. However, we acknowledge possible limitations in conducting the systematic literature review that may affect the analysis and interpretation of the results, such as heterogeneity in the types of interventions and outcome measures, which could hinder the feasibility of a meta-analysis and limit direct comparability between studies. This limitation will be addressed through rigorous analytical strategies and transparency in the presentation of findings.

Future research on digital health interventions for promoting health literacy among informal caregivers should adopt specific approaches to develop standardized frameworks for evaluating the effectiveness of these interventions, ensuring comparability and consistency in the results. Additionally, it is essential to design tailored solutions adapted to the diverse levels of digital literacy among informal caregivers, incorporating pilot testing to address their specific needs. Future studies should investigate strategies for integrating these technologies into healthcare systems, analyzing their sustainability and long-term impact. These actions could foster greater effectiveness and equity in the use of digital health technologies.

## Data Availability

The original contributions presented in the study are included in the article/Appendix A, further inquiries can be directed to the corresponding author.

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
