# Peer review of "The Effectiveness of eHealth Interventions in Promoting the Health Literacy of Informal Caregivers: A Systematic Literature Review Protocol"

_healthcare, 2024, doi:10.3390/healthcare12232354_

Round 1
Reviewer 1 Report
Comments and Suggestions for Authors
Dear Respectable Authors
Thank you for considering this important area of research related to eHealth interventions and their role in promoting health literacy. You intend to conduct a systematic review to investigate the effectiveness of eHealth interventions in informal caregivers. The methodology you used to conduct this research seems appropriate. However, some issues need to be corrected. I hope these comments are helpful and will increase your study consistency. My comments are as follows;
- Abstract, please add the tools you will use for quality appraisal. Please also add the main keywords and data items that you will extract from the included studies.
- Abstract, please add a statement that will show the application of the results of your study in a practice/ clinical setting.
- Introduction, please mention the reason for choosing this population for your study. Why do you focus on informal caregivers?
- Page 4, Material and Methods section, line 3, you stated that the PRISMA-P is available as a supplementary file, but there is not such a file in your supplementary materials.
- Please add a definition for informal caregivers or state what you mean by informal caregivers in this study.
- eHealth is a wide area of research and investigating its intervention is hard. Please add a clear definition for the intervention that you will investigate in your study.
- Please define how you measure the effectiveness of an eHealth intervention in promoting HL. Or what type of outcome do you investigate that approves that an intervention will promote HL?
- Subheading 2.1.5, line 2, the way you write this statement is a bit misleading. Please rephrase this statement (... excluded include ...).
-It's great that you use the ROB-2 checklist. But in my opinion, other types of studies will also be included in your study, and it will not be possible to evaluate their quality with this checklist. What plan do you have for evaluating the quality of other types of studies?
- If meta-analysis is possible, what outcomes will be investigated, and what type of effect sizes will be included?
- In my opinion, you can remove the heading 3 because such an item is not included in writing a protocol. Instead, in your discussion, explain a little more about the application of your results in practice, what kind of knowledge your results will cover/ gap of knowledge, and what possible research can be done based on these results in the future.
- Supplementary file, please use a better caption for this supplementary material. Also, you send a low-quality capture. It is better to add this material to a table. This table should include the following columns: Database; Search strategy/query; Number of records; and Date of search.
Cheers
Author Response
Por favor, veja o anexo

Reviewer 2 Report
Comments and Suggestions for Authors
Overall, the manuscript has issues with clarity and flow, which affect the readability of the text. It is recommended that the authors revise their writing style to improve the coherence of ideas and avoid overly short paragraphs that contain only a single sentence. An academic manuscript requires a more robust paragraph structure to allow for a thorough development of ideas.
The introduction should focus more on what is known and unknown about the study topic, providing a clearer context for the need to conduct this systematic review. Currently, it includes too much general information without focusing on the knowledge gaps the review seeks to address. It would be helpful to reorganize the introduction to make it more precise and aligned with the study's objectives.
The citation style contains inconsistencies, such as in the reference to the WHO's Global Digital Health Strategy 2020-2025. The citation format should be reviewed according to the journal's guidelines, avoiding unnecessary details such as "p. 11," which are typically not required in most modern citation styles.
The structure of the protocol does not seem to meet all the requirements of the PRISMA-P guidelines, which are essential for ensuring transparency and reproducibility in systematic reviews. The authors should review the PRISMA-P guidelines and ensure that all necessary elements are covered in the protocol.
There are discrepancies between what is registered in the PROSPERO protocol and what is described in the manuscript. This is a critical issue that must be addressed, as consistency between the two documents is fundamental to the study's credibility. It is recommended that the authors review both documents and ensure there are no differences in methods and objectives.
The search strategy presented for Scopus does not appear to comply with the PRESS guidelines that the authors mention. Given that the quality of the literature search is essential for a systematic review, it would be advisable for the authors to revise their search strategy to ensure it strictly follows the PRESS guideline recommendations.
The authors mention the inclusion of observational studies in the review but indicate they will use RoB-2 to assess bias, which is incorrect since this tool is specifically designed for randomized controlled trials. They should consider using appropriate tools for assessing bias in observational studies, such as ROBINS-I.
Although the authors have not yet presented results, it would be helpful to include examples of tables and figures to give a clear idea of how they plan to present the data from the included studies.
The discussion is minimal and does not contribute significantly to the manuscript's scientific value. It is recommended that the authors expand on the potential relevance of the expected results and how they may influence future research or clinical practice.
Comments on the Quality of English LanguageThe English could be improved to more clearly express the research.
Author Response
Please, see the attachment

Reviewer 3 Report
Comments and Suggestions for Authors
The protocol for systematic review is well described.
what is missing is the period of study inclusion.
Introduction is too long it is 2.5 pages and not in best flow of thoughts.
While many details are sufficiently described it looked bombarded.
Suggest summarise it in such flow: definitions --> why this topic is important --> challenges --> existing literature --> gaps to be addressed leading to the aim of this protocol.
For paragraph 10 of introduction, why this landscape of health is being described here? It may be relevant only if this systematic review is limited to EU. But if it is global and no restriction in geographical area, perhaps elaborate a few other examples to support the case globally.
2.1.4 outcomes
Authors only mentioned HLS as outcomes. How about Health Literacy Questionnaire (HLQ), Brief HL Screening (BHLS), sTOFHLA, REALM, etc.
2.2 Search strategy
one relevant online database. Since one is the mentioned, authors should just decide at this juncture to whether use Pubmed or Scopus.
2.4 Risk of bias
Results can be "low" or "high". could there be possibility that there is inadequate information to asses leading to need for category "unclear"?
2.5 coding
could be just transcribing or extracting and storing into Excel
is there new codes or nodes created?
3. Results
since most comparison will be post vs pre; it is possible to include more than one measurement tool than HLS.
4. Discussion can be improved by mentioning some of the well-known health interventions prior to SR. This can be appropriately inserted between the points/argument why you think this SR is important and what is the value preposition/aim of this SR.
4.1 Limitation
Authors are encouraged to also discuss what is the limitation expected prior to conducting the SR.
eg lack of agreed consensus on definition of eHealth or expecting high heterogeneity due to different outcomes measurements.
Comments on the Quality of English Language
English is okay.
Author Response
Please, see the attachment

Round 2
Reviewer 2 Report
Comments and Suggestions for Authors
The authors addressed the observations.
Author Response
Thank you very much for your revisions, which helped to improve the manuscript.
Reviewer 3 Report
Comments and Suggestions for Authors
Much of the content improved especially details in methodology.
One major concern is that the references are not in sequence and some reference are missing. So please check and update.
Author Response
Thank you very much for your revisions, which helped to improve the manuscript.
In accordance with your last suggestion, I have revised the manuscript again and identified that references 7 and 10 had not appeared in the manuscript by mistake. Thank you very much.